# Dreaming and Insomnia: Link between Physiological REM Parameters and Mentation Characteristics

**DOI:** 10.3390/brainsci10060378

**Published:** 2020-06-16

**Authors:** Fee Benz, Dieter Riemann, Bernd Feige

**Affiliations:** Department of Psychiatry and Psychotherapy, Medical Center-University of Freiburg, Faculty of Medicine, University of Freiburg, Hauptstraße 5, 79104 Freiburg, Germany; dieter.riemann@uniklinik-freiburg.de (D.R.); bernd.feige@uniklinik-freiburg.de (B.F.)

**Keywords:** insomnia disorder, dreaming, mental sleep activity, sleep, sleep state misperception, REM, awakening study, spectral power

## Abstract

(1) Background: An unresolved phenomenon of insomnia disorder is a discrepancy between objectively measured sleep and subjective complaints. It has been shown that rapid eye movement (REM) sleep might be especially vulnerable to an altered perception. The present work aimed to investigate the link between physiological REM parameters and mentation characteristics in REM sleep. (2) Methods: 22 patients with insomnia and 23 good sleepers indicating at least one REM mentation within an awakening study were included. Multivariate analyses of variance (MANOVAs) were calculated to examine group differences and effects of mentation characteristics on number of arousals, REM density, and spectral power prior to awakenings. (3) Results: Increased perceived wakefulness was related to lower delta, theta, and alpha power in the minute prior to the REM awakenings. Nevertheless, no group differences regarding spectral power were found. With respect to number of arousals and REM density, no significant effects of mentation characteristics and no group differences were found. (4) Conclusions: Our results suggest that spectral power in REM sleep is linked with altered sleep perception. Reduced delta, theta, and alpha power might be a signature of this modified REM sleep associated with a high level of perceived wakefulness. Future awakening studies are necessary to further explore the link between physiological REM parameters and sleep perception.

## 1. Introduction

Insomnia disorder is a highly prevalent disorder worldwide [1]. Approximately 10% suffer from its chronic form while 30–35% report insomnia symptoms, without meeting diagnostic criteria [2]. The disorder is characterized by a subjective report of having persistent difficulty with initiating/maintaining sleep or early morning awakening accompanied by daytime impairment [3]. Research has shown that insomnia disorder is associated with reduced quality of life [4] and an increased risk for developing cardiovascular disease [5] and mental disorders [6,7].

Although insomnia is a major public health problem, the pathophysiology is still not fully understood [1]. An unresolved common finding remains the so called “misperception of sleep”, i.e., a discrepancy between objective measured sleep through polysomnography and the subjective complaints [8]. Many patients with insomnia disorder tend to underestimate their total sleep time and overestimate the time they are spending awake in bed [9,10,11]. Current pathophysiological models emphasize a heightened arousal in the cognitive, emotional, and physiological domains during night and day in patients with insomnia disorder that may help to explain the phenomenon of misperception [12,13,14].

Rapid eye movement (REM) sleep, as the most aroused sleep state, may play a central role for the subjective experience of insomnia [11,14]. Besides rapid eye movements, this unique sleep state is characterized by a desynchronized electroencephalogram (EEG) with theta and alpha waves, muscle atonia, and vivid dreaming [15]. Feige and colleagues [11] investigated a dataset of 100 patients with primary insomnia and 100 age and sex-matched healthy sleepers and found the amount of REM sleep to predict the amount of subjective wake time in the patient group as well as increased arousal densities specifically during REM sleep in insomnia patients. This modified REM sleep quality or “REM sleep instability” may contribute to the experience of disrupted sleep and the often found discrepancy between subjective and objective sleep parameters [14].

In order to gain more direct insight into the characteristics of this modified REM sleep, awakening studies are needed for assessing sleep mentation and perception reports for defined sleep stages [16,17]. However, systematic research in this area is scarce. Schredl summarized the existing literature on dreams in patients with sleep disorders [18]. The few studies on sleep mentation in insomnia patients suggest that their dreams reflect current stressors and wake-life concerns which is in line with the continuity hypothesis formulated by Hall and Nordby [19]. In a study using morning dream recall, Schredl and colleagues [16] found that dream content of insomnia patients was characterized by more negative emotions and more negative elements in comparison to the dream content of good sleeper controls. Another method of data collection comprises in-laboratory awakening studies which offer a more direct approach to study sleep perception and dream content. To the best of our knowledge, only three studies investigated dream content of insomnia patients using awakening studies. Ermann [20] conducted a REM awakening study and found that dreams of insomnia patients more often contained negatives in self-description and negatives in the dream description. Perussé and colleagues [17] found that dreams of insomnia patients elicited after REM sleep awakenings were characterized by more negative elements than positive ones. Furthermore, the negative dream content was associated with lower sleep efficiencies in insomnia patients. Finally, Feige and colleagues [21] have shown that patients with insomnia disorder rated their REM sleep mentation as more negatively toned compared to good sleeper controls. They also conducted sleep stage 2 (N2) awakenings, where no significant group difference for negative feelings was observed. Compared to healthy sleepers, patients with insomnia disorder additionally reported having been awake after awakenings from REM sleep more frequently but not from N2, and they were less sure when indicating they had been asleep. These findings also support the specific role of REM sleep for the subjective experience of insomnia.

The current paper presents additional data collected within the awakening study by Feige et al. [21]. Due to the specific role of REM sleep and the previous finding of Feige and colleagues [21] that sleep mentation was rated more negatively by insomnia patients compared to good sleepers specifically during REM awakenings, we aimed to investigate the physiological correlates of mentation characteristics during REM sleep. To do this, we examined the sleep EEG preceding experimental REM awakenings in patients with insomnia disorder and good sleeper controls. This work expands upon the study by Feige et al. [21] by connecting physiological parameters of REM sleep with dream characteristics in this sleep stage. More specifically, we investigated the link between physiological parameters of REM sleep (number of arousals, REM density, and spectral analysis) and characteristics of the REM sleep mentation (clarity, control, visuality, emotional valence, and a measure of perceived wakefulness). Therefore, only those participants reporting at least one REM sleep mentation were included in our analysis. To our knowledge, this is the first analysis investigating the link between physiological REM parameters and mentation characteristics reported after experimental awakenings in insomnia patients. Thus, the present work aims at contributing new insights to the existing limited literature on dreaming in insomnia.

With respect to the REM instability hypothesis, we hypothesized that an increased number of arousals is linked to higher perceived wakefulness and more wake-like mentation (less visual but more thought-like). Since REM density and spectral power can be considered as measures of the intensity of REM sleep, we hypothesized that higher values of both variables are linked with decreased perceived wakefulness and more visual mentation. This hypothesis is based on the finding that dream reports are more vivid during phasic REM sleep compared to tonic REM sleep [22,23]. During phasic periods, bursts of rapid eye movements occur while no rapid eye movements occur in tonic periods [23]. Hence, it might be easier for the dreamer to identify this mentation as a dream and to correctly discriminate between wakefulness and sleep.

## 2. Materials and Methods

### 2.1. Design

Details of the current case-control study are described by Feige and colleagues [21]. The study protocol of the primary study was approved by the local ethics committee of the University of Freiburg Medical Centre (Vote 399/12; 15 October 2012) and the study was conducted in accordance with the Declaration of Helsinki. The research protocol included four consecutive nights with polysomnography (PSG) recordings in the sleep laboratory of our department, which has been accredited by the German Society for Sleep Research and Sleep Medicine [24]. The first night was the adaptation night and screened for sleep apneas and periodic leg movements. The second night served as baseline night gaining information about the amount of difference between objective (PSG) and subjective (sleep diary) measured total sleep time (TST). In the third and fourth nights, three experimental awakenings occurred in each night, in randomized order from REM sleep in one night and from stage N2 in the other.

### 2.2. Participants

In this study, patients were recruited through the database of former inpatients with insomnia and the outpatient sleep disorder clinic while good sleeper controls (GSC) were recruited by word of mouth. The screening procedure included an extensive physical and psychiatric investigation, a routine blood sample testing, a urine sample including drug screening, the Structured Clinical Interview for DSM-IV, Axis I (SCID-I [25]), and the sleep diaries of the “Deutsche Gesellschaft für Schlafforschung und Schlafmedizin” (DGSM, http://www.dgsm.de/) for two weeks. In total, 42 GSC and 41 patients with insomnia were screened in order to reach 28 participants per group that were required based on the sample size calculation (for details see Feige et al. [21]). Before the screening process, all participants were informed in detail about the study’s purpose, design, and potential risks. In addition, they were informed that participation is voluntary and that the informed consent can be withdrawn at any time without giving a reason. For the whole study participation, participants were reimbursed with 300€.

Inclusion criteria for the study were: (1) age 25 to 65 years, (2) normal hearing, and (3) signed informed consent. For the patient group, additional inclusion criteria was required: (4) insomnia disorder according to DSM 5 or primary insomnia according to DSM IV and (5) underestimation of total sleep time of at least 60 min based on a discrepancy between subjective (14-day sleep diary) and objective TST (PSG derived TST of the baseline night) in order to guarantee a certain degree of misperception. Exclusion criteria were: (1) acute or lifetime psychiatric disorders (evaluated with the SCID-I) with the exception of insomnia for the patient group, (2) other sleep disorders apart from insomnia in the patient group (evaluated by clinical interview and results from the PSG of the first night in the sleep laboratory), (3) shift work or transmeridian flights within the last four weeks and irregular sleep-wake rhythms (frequent shifts of bed times > 1 h), (4) regular intake of any psychotropic substance influencing sleep within the two weeks before as well as during study participation, (5) ongoing psychotherapy, (6) pregnancy or lactation, (7) clinically significant, severe, or unstable medical diseases that have an impact on sleep, and (8) participants with an Apnea Index > 5 per h/a periodic leg movements in sleep (PLMS) with arousal index > 5 per hour during the first night of the study.

### 2.3. Questionnaires

In addition to the screening procedure, the following questionnaires were filled out by participants in order to describe both samples: Insomnia Severity Index (ISI [26]), Pittsburgh Sleep Quality Index (PSQI [27]), Epworth Sleepiness Scale (ESS [28]), Glasgow Sleep Effort Scale (GSES [29]), Ford Insomnia Response to Stress Test (FIRST [30,31]), Dysfunctional Beliefs and Attitudes about Sleep (DBAS [32]), Pre-Sleep Arousal Scale (PSAS [33]), Beck Depression Inventory (BDI [34]), and State Trait Anxiety Inventory (STAI [35]).

### 2.4. PSG and EEG Analyses

PSG was recorded using 24-channel Somnomedics PSG for eight hours, from 11 pm until 7 am. Experienced raters who were blind to participant status (patient vs. GSC) scored PSGs visually in 30 s epochs according to the criteria of Rechtschaffen and Kales [36], comprising the modifications by the AASM [37]. During the first and second nights abdominal and thoracic effort, nasal airflow, oximetry, and bilateral tibialis anterior electromyography (EMG) were monitored. Sleep EEG included C3/A2 and C4/A1 derivations filtered with a time constant of 0.3 s and a low-pass at 75 Hz and then digitized at 256 Hz. Sleep continuity parameters included TST, sleep efficiency index (SEI; ratio of TST to time in bed), sleep onset latency (SOL; time from lights out until the first occurrence of sleep stage N2), and the number of wake periods (NWP). Sleep architecture variables included the percentages of stages Wake, N1, N2, N3, and REM within sleep period time. REM sleep variables included REM latency (interval from sleep onset to the occurrence of the first REM period) and REM density (calculated as in previous studies, e.g., [11]). Microarousal analysis was conducted according to ASDA criteria [38] and previous work of our group [11]. Spectral analysis was performed on EEG channel C3-A2 by averaging logarithmic spectral power obtained from all artefact-free 5s epochs within the 60s interval preceding the first tone of each REM sleep awakening (cf. [39,40]). Spectral power values were summed within the following frequency bands: Delta 0.5–3.5 Hz, theta 3.5–8 Hz, alpha 8–12 Hz, sigma 12–16 Hz, beta 16–32 Hz, and gamma 32–48 Hz.

### 2.5. Awakening Procedure

In each experimental night (nights three and four), participants were awoken three times, in one night after five minutes of consolidated stage N2 and in the other night after five minutes of consolidated REM sleep, in a randomized order. Experimental awakenings started with the second NREM-REM cycle and were conducted by raters trained in sleep EEG scoring. Participants were awoken through an auditory stimulus, a 500 ms white noise sound presented through a speaker in the room with increasing intensity every 4 s, in steps of 1 dB starting at 5 dB hearing level. Awakenings were indicated through a double click of a microswitch that was affixed to the thumb of the dominant hand of the participant. Double clicking was chosen in order to discriminate between intended and unintended depressions of the microswitch. After the experimenter started the auditory stimulation and the participant indicated to be awake the standardized interview started. This interview was based on questions on sleep-wake discrimination adapted from Weigand and colleagues [41] and on questions about sleep mentation by Schredl and colleagues [16] and included: (1) judgement on the state prior to the auditory stimulus (awake or asleep; 0–1), (2) certainty of this judgement (not sure, quite sure, very sure; 0–2), (3) presence of mentation (was something on your mind before you heard the tone; no–yes; 0–1), (4) three characteristics of this mentation rated as binary choices: Clarity (vague or clear), visuality (thought- or image-like), and control (mentation just happened or was self-controlled), (5) presence of positive emotions (none, some, moderate, strong; 0–3), and (6) presence of negative emotions (none, some, moderate, strong; 0–3). During the interview, the experimenter came into the room and turned on a little bedside lamp, the main room lights remained turned off. Prior to lights out on both experimental nights, participants were familiarized with the whole awakening procedure.

### 2.6. Statistical Analyses

A new variable was created by combining the first two items of the awakening interview (having been awake or asleep and certainty of this judgement), forming an index of how awake subjects judged their state prior to the awakening. This variable was labeled “perceived wakefulness” with values on a scale from one (no perceived wakefulness) to six (high perceived wakefulness): Asleep and very sure (one), asleep and quite sure (two), asleep and not sure (three), awake and not sure (four), awake and quite sure (five), and awake and very sure (six). Since all awakenings occurred from polysomnographically ensured REM sleep, “perceived wakefulness” is an awakening-based measure of sleep misperception. All participants who indicated at least one REM mentation at night were entered in the analyses (see below). For each participant, we averaged the measures of the standardized interview obtained from all REM awakenings and computed a mean for each physiological parameter across a period of one minute prior to the first tone of REM awakenings. The one-minute time frame was used since we were interested in physiological correlates of mentation characteristics and thus, we decided to choose a timeframe close to the awakenings. Previous studies used time frames between 16 s and 120 s (as cited in [42]). In the awakening study by Wittmann et al. [42] investigating the association between dream recall and spectral power in a small sample of healthy females, a time window of 32 s was used. The authors discussed that their time window might have been too short to find significant associations. Therefore, we chose a time frame of one minute prior to awakenings.

Group differences and effects of mentation characteristics were assessed using multivariate analyses of variance (MANOVAs) with factor group and the following covariates: Age, mentation clarity, mentation visuality, mentation control, positive feelings, negative feelings, and perceived wakefulness. Two separate MANOVAs were conducted, one for arousal and REM density one minute before awakenings and one for spectral power (delta, theta, alpha, sigma, beta, and gamma) one minute before awakenings. Multivariate statistics were based on Wilk’s Lambda. Further univariate analyses were performed to explore the relation between mentation characteristics and REM parameters. MANOVAs were computed to avoid alpha error inflation. Thus, univariate results are only considered where the corresponding multivariate test was significant. *p*-Values of < 0.05 were considered as statistically significant, and < 0.1 as marginally significant or tendencies.

Descriptive data are reported as means and standard deviations (SD). Statistical analyses were performed using the software R (version 3.6.2, The R Foundation for Statistical Computing 2019, https://www.r-project.org/).

## 3. Results

### 3.1. Sample Description

Including only participants who indicated at least one mentation in the REM awakening night resulted in a subgroup of *n* = 45 participants (22 patients with insomnia disorder, 23 GSC) in comparison to the original sample of *n* = 54 participants (27 per group) reported by Feige and colleagues [21]. Sample characteristics are shown in Table 1. The groups did not differ with respect to age and had comparable gender proportions (63.3% females in patient group, 56.5% females in control group). On average, the patient group suffered from insomnia for 12.53 years representing a group with high chronicity. Accordingly, all insomnia specific measures (ISI, ESS, GSES, FIRST, PSAS, DBAS-16, and PSQI) were significantly increased in the patient group. Likewise, sleep diary data were significantly altered in insomnia patients, apart from time in bed, duration of daytime naps and exhaustion in the evening (marginally significant). The subjective TST according to the sleep diary was 95.74 min shorter in the patient group compared to the control group. Furthermore, the patient group showed significantly increased values of depression and anxiety (BDI, STAI), however within a subclinical range.

### 3.2. Number of Arousals and REM Density

Table 2 shows number of arousals and REM density one minute before awakening (both variables averaged over all REM awakenings) together with results from multi- and univariate ANOVA tests for factor group and all covariates (age, mentation clarity, mentation visuality, mentation control, positive feelings, negative feelings, and perceived wakefulness of sleep). No significant effects on mean number of arousals and mean REM density were reached in the multivariate analysis. The groups did not differ with respect to number of arousals and REM density and there were no significant associations between arousals and REM density and the covariates age and mentation characteristics.

### 3.3. Spectral Analysis

Table 3 shows multi- and univariate ANOVA results for sleep EEG spectral power one minute before awakening (averaged over all REM awakenings). Using Wilk’s Lambda statistic, there was a significant multivariate effect of mean perceived wakefulness on mean spectral power, Λ = 0.64, F(6, 30) = 2.86, *p* < 0.05. Increased perceived wakefulness was related to significantly reduced delta, theta, and alpha power. Sigma, beta, and gamma band power did not show significant associations with perceived wakefulness. No group differences could be detected and there were no significant associations between spectral power and the other covariates examined. Multicollinearity tests revealed two correlations above 0.9: Correlation between sigma and alpha as well as correlation between sigma and beta. Thus, the analysis was also performed without sigma to exclude possibilities of spurious relationships or wrong variance estimation that can happen in this case. Again, there was a significant multivariate effect of mean perceived wakefulness on mean spectral power, Λ = 0.65, F(5, 31) = 3.32, *p* < 0.05. Since results did not differ, we chose to report the full table including sigma.

Figure 1 illustrates the main effect of “perceived wakefulness” on spectral power. It shows the full log power spectra, across both groups, for subjects below and above the median “perceived wakefulness” value of 1.5. The perceived wakefulness-related differences mainly occur within delta, theta, and lower alpha band.

## 4. Discussion

To our knowledge, this is the first analysis that investigated the relation between different sleep mentation characteristics using experimental awakenings from REM sleep and possible physiological correlates (number of arousals, REM density, and spectral power prior to the awakenings) in a sample of insomnia patients with a typical degree of sleep misperception and good sleepers. The current work presented additional data collected within the awakening study by Feige and colleagues [21] aiming to provide new insights into the specific role of REM sleep for the subjective experience of insomnia.

Based on the “REM instability hypothesis” [14], we hypothesized that an increased number of arousals would be linked with higher perceived wakefulness and less visual, but more thought-like sleep mentation. Furthermore, we hypothesized that increased REM density and increased spectral power are related with reduced perceived wakefulness and more visual mentation.

The first hypothesis was not supported by our results. Insomnia patients and good sleeper controls did not differ in number of arousals in REM sleep prior to awakenings and no relation between the number of arousals and mentation characteristics was found. One explanation for this finding might be that the analyzed epoch of 60 s prior to the awakening might have been too short, resulting in low resolution of this variable. However, also the main study of Feige et al. [21] did not find group differences in either NREM or REM sleep. We are not aware of a similar study examining the link between number of arousals and sleep perception directly measured through an awakening design. Thus, it would be interesting to investigate the potential link of number of arousals and sleep mentation in future studies with larger sample sizes.

Furthermore, no association between REM density prior to the awakenings and sleep mentation characteristics was found. A pilot study by Wehrle and colleagues [43] combining simultaneous functional magnetic resonance imaging (fMRI) and polysomnographic recordings in human REM sleep, showed that rapid eye movements were associated with an activation of geniculate bodies and the secondary visual cortex. Their finding suggests that ponto-geniculo-occipital (PGO) waves, found in invasive animal models and hypothesized to initiate and maintain the vivid visual experience during REM sleep [43,44], are also present in human REM sleep. In a later study by Wehrle et al. [45] using the same approach, they found a lack of reactivity to acoustic stimulation in phasic REM sleep compared to tonic REM sleep suggesting that phasic REM sleep might be less vulnerable to be perceived as awake. Although a relation between higher REM density and more visual mentation as well as less perceived wakefulness seems conceivable, our data did not support this association. Future PSG studies using an awakening design should further explore the association between REM density and mentation characteristics like visuality and perceived wakefulness.

Our last hypothesis about a link between perceived wakefulness and spectral power was supported by our results. Reduced perceived wakefulness was related to increased delta, theta, and alpha power in the minute prior to the REM awakenings. For the other spectral power bands, no significant associations were found. Moreover, spectral power prior to the awakenings did not differ between the groups and was not related to the other mentation characteristics examined (clarity, control, visuality, and emotional valence of REM sleep mentation).

One might speculate that increased delta power might be related to eye movement artifact (electrooculogram, EOG). However, REM density did not differ between the groups and no significant associations were found on this variable; also, the differences similarly extended into the theta and alpha bands, which are unlikely to originate from the EOG. Therefore, the obtained spectral power differences probably represent cortical activity.

Studies investigating spectral power in REM sleep in insomnia are scarce and results are heterogenous [46,47,48]. Besides differences in study designs, some authors examined absolute spectral power measuring the actual power in frequency bands while others focused on relative spectral power (the power in a frequency band divided by total power across all frequency bands). One advantage of relative spectral power is a reduction of variance between subjects caused by individual differences. However, by calculating relative power, spectral power bands are not independent from one another. Hence, it is hard to interpret changes in specific frequency bands, as an increase e.g., in relative high-frequency activity may be due to a decrease in low-frequency activity, which dominates total power.

Merica and colleagues [46] examined spectral characteristics of sleep EEG for the first four REM and NREM periods in 20 chronic insomnia patients and 19 healthy subjects. In REM sleep, they found lower spectral activity in the delta and theta bands and increased power in the faster frequency bands. St. Jean and colleagues [47] investigated absolute and relative spectral power on two consecutive in-laboratory nights in a sample of 21 good sleepers and 46 insomnia patients subtyped into paradoxical (*n* = 20) and psychophysiological (*n* = 26) insomnia. With respect to REM sleep, they found lower relative activity in slower frequency bands in the paradoxical insomnia group compared to the good sleeper controls as well as less relative theta activity in the psychophysiological insomnia group compared to the control group. Krystal and colleagues [48] compared EEG frequency spectra from REM and NREM sleep from one night of ambulatory PSG of 20 good sleepers and 30 insomnia patients. Patients were subtyped in a group of objective insomnia (*n* = 18) and subjective insomnia (*n* = 12) who were characterized by an underestimation of total sleep time compared with PSG. They did not find differences in REM spectral activity between the groups.

In comparison to the studies of Merica et al. [46] and St. Jean et al. [47], we found no group differences in spectral power when simultaneously considering perceived wakefulness. However, supposing that the insomnia group of Merica et al. [46] was characterized by increased misperception as is typically found for patients with insomnia, the group difference in their study could have resulted from the perceived wakefulness-related spectral signature found in the current study. Similarly, the finding of St. Jean et al. [47] that paradoxical insomnia patients showed lower relative activity in slower frequency bands is in accordance with our result that participants with a higher degree of perceived wakefulness indicated lower spectral power mainly in the delta, theta, and lower alpha frequency bands.

Merica et al. [46] and St. Jean et al. [47] interpret reduced delta power in REM sleep in terms of the hyperarousal concept of insomnia [12,13]. However, while delta power in NREM sleep is an accepted correlate of sleep depth, the same is not true for REM sleep. As activated brain state postulated to subserve important functions in reappraisal and emotion regulation [15,49,50], spectral power changes in REM sleep can be expected to reflect properties of sleep mentation in addition to “intensity” or “depth” of REM sleep. Defining the latter properties has proven difficult, however. REM density is an accepted correlate of REM sleep disinhibition at least in depression [51] and awakening thresholds have been reported to be increased in phasic REM episodes in some studies [52] but not in others [21]. These concepts of REM sleep intensity are quite different, though, as it is conceivable that a disinhibited REM sleep can be at the same time particularly “active” and well protected from external disturbance. Feige and colleagues [21] did not find lowered awakening thresholds or REM density differences in REM sleep in insomnia patients, but upon awakening from REM sleep the patients reported, significantly more often, having been awake. This indicates that a different aspect of REM sleep quality is modified in insomnia [11,14]. Our current results point toward the interpretation of reduced delta, theta, and alpha power as a signature of this modified REM sleep associated with a high level of perceived wakefulness.

Besides many strengths of the present awakening study, the current work also has some limitations. First, the sample comprised a subsample of the study by Feige et al. [21], because only those participants indicating at least one mentation in the REM awakening night were included in the analysis. Hence, future studies should dedicate to this topic using larger sample sizes to increase statistical power. Since this study excluded participants with psychiatric comorbidity, it would be interesting for future studies to investigate the link between physiological parameters and mentation characteristics in individuals with greater levels of psychopathology and/or reported disturbed dreaming. Furthermore, it cannot be ruled out that the laboratory setting might have affected mentation characteristics, despite the habituation nights. However, this methodological design is necessary to address the research question of this study. In addition, activity of the different frequency bands might fluctuate between nights [47] suggesting that future studies should investigate more nights instead of one night. Another limitation refers to the fact that we did not assess usual dream recall frequency of participants. There is some evidence of differences in mentation reports based on whether the participants are habitual dream recallers or not [53]. Future research could investigate whether associations between physiological REM parameters and mentation characteristics differ dependent on usual dream recall frequency. Finally, mentation characteristics as well as physiological parameters from all REM awakenings were averaged. It would be interesting to analyze intraindividual differences and to see if effects may differ in the course of the night. Because of lack of data due the fact that many participants did not indicate a mentation per awakening, we decided to focus on between-subjects effects model. Thus, future studies might benefit from conducting more than three awakenings per night to get more intraindividual data.

## 5. Conclusions

To conclude, the present work showed a significant relation between increased spectral power in the delta, theta, and lower alpha frequency bands and reduced sleep perceived wakefulness as measured with an awakening study providing a direct approach to investigate perception of sleep. Participants (patients with insomnia disorder and good sleeper controls) with a higher degree of perceived wakefulness showed significantly reduced log spectral power in these frequency bands. Feige et al. [21] found that insomnia patients are more likely to perceive REM sleep as wakefulness indicating that patients may dream to be awake. Our results suggest that spectral power in REM sleep might be linked with this altered perception. Future awakening studies are necessary to further examine the possible link between spectral power in REM sleep and hyperarousal in insomnia patients. Since REM sleep is important for the subjective experience of insomnia and vulnerable to modifications of consciousness, the investigation of physiological correlates of dreaming and mentation activity in insomnia is a promising research field to better understand the pathophysiology of the disorder.

## Figures and Tables

**Figure 1 brainsci-10-00378-f001:**
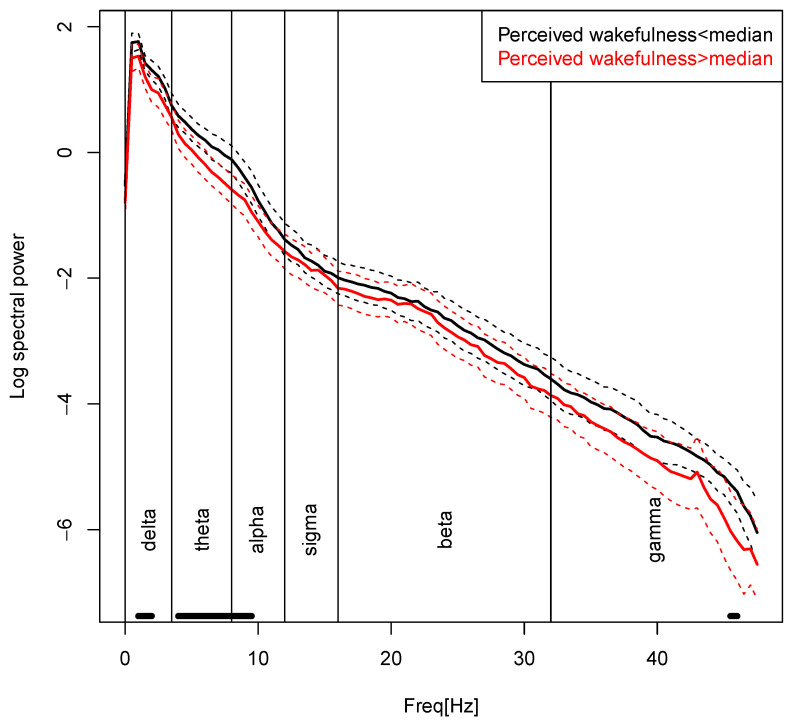
EEG spectral power for the main effect “perceived wakefulness”. Average log power spectra and frequency bin-wise 95% confidence intervals are shown for subjects (both patients with insomnia disorder (ID) and good sleeper controls (GSC)) with perceived wakefulness values below (black lines) and above (red lines) the median value of 1.5. Frequency band boundaries are indicated by vertical black lines, and a thick black bar just above the frequency axis marks frequencies with uncorrected bin-wise group difference *p* < 0.05.

**Table 1 brainsci-10-00378-t001:** Sample characteristics.

	M	F	M	F		
*n*	10	13	8	14		
	Mean ± SD		Mean ± SD		*t*	*p*
Age	43.09 ± 11.47		44.32 ± 11.87		0.35	0.725
Duration of insomnia (years)	0.00 ± 0.00		12.53 ± 16.07		3.66	0.001
BDI	1.30 ± 2.03		9.36 ± 7.03		5.17	0.000
STAI	27.83 ± 5.13		39.19 ± 9.67		4.80	0.000
ISI	1.96 ± 1.94		16.27 ± 4.23		14.47	0.000
ESS	4.35 ± 2.59		7.86 ± 4.11		3.42	0.002
GSES	0.48 ± 0.85		6.36 ± 3.50		7.68	0.000
FIRST	17.83 ± 4.13		24.95 ± 5.35		4.99	0.000
DBAS-16	29.09 ± 17.13		73.91 ± 22.52		7.49	0.000
PSAS-SA	8.74 ± 1.25		12.45 ± 3.66		4.51	0.000
PSAS-CA	10.57 ± 2.79		19.73 ± 7.90		5.14	0.000
PSQI	2.57 ± 1.47		10.68 ± 3.50		10.07	0.000
Sleep diary						
TIB	483.50 ± 47.33		504.41 ± 59.08		1.30	0.203
TST	439.44 ± 51.04		343.70 ± 57.21		−5.86	0.000
SEI	90.84 ± 5.32		68.78 ± 12.90		−7.42	0.000
SOL	10.75 ± 4.56		42.52 ± 31.02		4.75	0.000
WAKE	8.14 ± 8.53		48.77 ± 46.71		4.01	0.001
EMA	25.18 ± 21.99		69.42 ± 50.82		3.75	0.001
NAP	5.89 ± 6.41		8.55 ± 21.52		0.56	0.583
MOOD_EVE	4.83 ± 0.94		3.89 ± 0.91		−3.37	0.002
PERFORMANCE_EVE	2.00 ± 0.76		3.00 ± 0.62		4.78	0.000
EXHAUSTION_EVE	1.65 ± 0.42		1.93 ± 0.51		1.98	0.055
RECOVERY_MOR	1.80 ± 0.56		3.02 ± 0.63		6.81	0.000
MOOD_MOR	4.95 ± 0.83		3.69 ± 0.78		−5.18	0.000

GSC = Good Sleeper Controls; ID = Patients with Insomnia Disorder; M = Male; F = Female; BDI = Beck Depression Inventory; STAI = State Trait Anxiety Inventory, trait version; ISI = Insomnia Severity Index; ESS = Epworth Sleepiness Scale; GSES = Glasgow Sleep Effort Scale; FIRST = Ford Insomnia Response to Stress Test; DBAS-16 = Dysfunctional Beliefs and Attitude about Sleep-16 items; PSAS = Pre-Sleep Arousal Scale; SA = Somatic Arousal; CA = Cognitive Arousal; PSQI = Pittsburgh Sleep Quality Index; The following measures are based on sleep diary data: SOL = Sleep Onset Latency; TST = Total Sleep Time; SEI = Sleep Efficiency Index; TIB = Time in Bed; EMA = Early Morning Awakening; WAKE = Total time spent awake at night; NAP = Duration of daytime nap; MOOD_EVE = Mood in the evening; PERFORMANCE_EVE = Daytime performance rated in the evening; EXHAUSTION_EVE = Exhaustion in the evening; RECOVERY_MOR = Recovery at morning; MOOD_MOR = Mood in the morning.

**Table 2 brainsci-10-00378-t002:** MANOVA results for the REM parameters number of arousals and REM density one minute before awakening (averaged over all REM awakenings). The top line shows the multivariate statistics. For all covariates, the effect estimates are given as change per unit (B).

	GSC (*n* = 23) Mean ± SD	Patients with ID (*n* = 22) Mean ± SD	Group			Age			Mentation Clarity			Mentation Visuality			Mentation Control			Positive Feelings			Negative Feelings			Perceived Wakefulness	
			*F*	*p*	*B*	*F*	*p*	*B*	*F*	*p*	*B*	*F*	*p*	*B*	*F*	*p*	*B*	*F*	*p*	*B*	*F*	*p*	*B*	*F*	*p*
Multivariate analysis (Wilk’s Lambda)			0.98	0.686		0.98	0.652		0.94	0.346		0.94	0.334		0.99	0.792		0.93	0.261		0.96	0.489		0.88	0.116
Number of arousals	0.25 ± 0.27	0.23 ± 0.36	0.04	0.844	0.00	0.01	0.906	0.01	1.20	0.280	−0.19	0.15	0.701	−0.06	0.00	0.973	0.06	0.26	0.615	0.12	0.91	0.347	−0.14	4.68	0.037
REM density (%)	7.28 ± 3.77	8.17 ± 3.36	0.78	0.383	−0.04	0.74	0.395	0.08	0.43	0.515	3.48	1.69	0.202	1.19	0.43	0.517	0.87	1.95	0.171	−1.28	1.04	0.316	0.60	0.65	0.427

**Table 3 brainsci-10-00378-t003:** MANOVA results for spectral power one minute before awakening (averaged over all REM awakenings). The top line shows the multivariate statistics. For all covariates, the effect estimates are given as change per unit (B).

	GSC (*n* = 23) Mean ± SD	Patients with ID (*n* = 21) Mean ± SD	Group			Age			Mentation Clarity			Mentation Visuality			Mentation Control			Positive Feelings			Negative Feelings			Perceived Wakefulness	
			*F*	*p*	*B*	*F*	*p*	*B*	*F*	*p*	*B*	*F*	*p*	*B*	*F*	*p*	*B*	*F*	*p*	*B*	*F*	*p*	*B*	*F*	*p*
**Multivariate analysis** **(Wilk’s Lambda)**			0.92	0.863		0.75	0.171		0.88	0.688		0.89	0.734		0.97	0.990		0.92	0.861		0.83	0.412		0.64	0.025
delta 0.1–3.5 Hz	3.30 ± 0.35	3.22 ± 0.37	0.69	0.413	0.00	0.26	0.616	−0.11	0.66	0.422	−0.14	0.31	0.581	−0.04	0.00	0.981	−0.09	1.78	0.191	−0.07	0.45	0.509	−0.18	4.96	0.032
theta 3.5–8 Hz	2.36 ± 0.46	2.27 ± 0.51	0.44	0.513	0.01	0.49	0.490	−0.19	0.59	0.448	−0.15	0.99	0.327	−0.12	0.03	0.860	−0.11	1.66	0.206	0.04	0.02	0.885	−0.24	5.00	0.032
alpha 8–12 Hz	1.26 ± 0.58	1.24 ± 0.64	0.06	0.810	0.01	2.32	0.137	−0.08	0.08	0.779	−0.16	1.33	0.257	−0.09	0.00	0.950	−0.15	2.01	0.165	−0.14	0.79	0.381	−0.35	7.59	0.009
sigma 12–16 Hz	0.25 ± 0.68	0.31 ± 0.57	0.03	0.864	0.02	5.61	0.024	−0.14	0.62	0.437	−0.10	0.22	0.641	−0.01	0.00	0.986	−0.15	1.44	0.238	−0.18	0.95	0.336	−0.20	2.26	0.142
beta 16–32 Hz	0.83 ± 0.76	0.86 ± 0.63	0.00	0.982	0.02	7.00	0.012	0.01	0.06	0.810	0.13	0.32	0.573	0.14	0.07	0.795	−0.16	1.17	0.287	−0.25	1.35	0.253	−0.10	0.41	0.524
gamma 32–48 Hz	−1.01 ± 0.69	−1.02 ± 0.86	0.03	0.865	0.02	2.58	0.118	−0.11	0.14	0.710	0.21	0.54	0.468	0.30	0.44	0.512	−0.15	0.74	0.396	−0.36	2.04	0.162	−0.14	0.68	0.416

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
