# Peer review of "Dreaming and Insomnia: Link between Physiological REM Parameters and Mentation Characteristics"

_brainsci, 2020, doi:10.3390/brainsci10060378_

Round 1
Reviewer 1 Report
In the paper “Dreaming and insomnia: Link between physiological REM parameters and mentation characteristics” the authors investigates the link between physiological REM parameters and mentation characteristics in patients with insomnia and in good sleepers. They found that reduced delta, theta and alpha spectral power in REM sleep is linked with altered sleep perception.
The paper is clear and well-written. The applied methodology is rigorous and the conclusions are supported by the results.
Author Response
We thank the reviewer for the appreciation of our efforts!
Reviewer 2 Report
Thank you for the opportunity to review this manuscript, which details an in-laboratory awakening study assessing physiological REM sleep parameters and sleep-state perception by insomnia diagnostic status in sleepers who reported at least one dream mentation. The unique experimental design provides an opportunity to further examine the processes of sleep perception and REM sleep characteristics, and the methodology is rigorous. Overall, the manuscript is well-written and provides additional information on how spectral power in REM sleep can be used to better understand the altered perception in insomnia. The main area of improvement could be clarity of the research question and how this study specifically expands on what was previously reported by Feige et al., 2018. Additional points discussed below.
Introduction:
- As mentioned above, it would be helpful to present the findings of the Feige et al., paper and explain how the present study will expand upon or differs from those results in the introduction.
- Potentially connecting to the point above, it was not until the Methods that it was explained that only participants with at least one reported REM sleep mentation were used for the analyses. Stating this choice and providing the rationale for it when discussing the purpose of the present study might address questions on how this study differs from that previously published.
Methods:
- Was the auditory stimulus presented through a speaker in the room or through headphones?
- What was the rationale for the one minute timeframe for computing the mean for the physiological parameters, as opposed to another length of time?
- In addition to other measures of assumptions, was the assumption of multicollinearity examined prior to conducting the MANOVAs?
Results:
- In Table 1, it is unclear what the following variables represent and what questionnaire was used to capture them: MOOD_EVE, PERFORMANCE_EVE, EXHAUSTION_EVE, RECOVERY_MOR, &
- The end of Table 2 & 3 were cut-off, so could not evaluate them as a whole. It looks like there were significant differences among the spectral analysis for age but was not mentioned in text.
Discussion/Limitations:
- Given that the perceived wakefulness was found in both groups, the authors may consider tempering some of their statements in the discussion that suggest their results may be linked to the altered perception in insomnia and I am wondering whether the title should be changed as it gives the impression that there were links between REM parameters the mentation characteristics.
- Some considerations that may be discussed in limitations or future research are:
- There is some evidence of differences on mentation reports based on whether the participants are habitual dream recallers. Was a baseline assessment given regarding dream recall in these participants.
- Despite the habituation nights, the laboratory setting may impact the mentation characteristics.
- The sample was comprised of a rather “clean” sample with little complexity and psychiatric symptoms. It would be interesting to see this work expanded to individuals with greater levels of psychopathology and/or reported disturbed dreaming.
